# Chinese College Students’ Physical-Exercise Behavior, Negative Emotions, and Their Correlation during the COVID-19 Outbreak

**DOI:** 10.3390/ijerph191610344

**Published:** 2022-08-19

**Authors:** Shan-Shan Han, Bo Li, You-Zhi Ke, Guang-Xu Wang, Shu-Qiao Meng, Ya-Xing Li, Zhong-Lei Cui, Wen-Xia Tong

**Affiliations:** 1Institute of Sports Science, Nantong University, Nantong 226019, China; 2School of Physical Education, Shanghai University of Sport, Shanghai 200438, China; 3College of Physical Education, Henan Normal University, Xinxiang 453007, China; 4Physical Education College, Yangzhou University, Yangzhou 225127, China; 5Physical Education College, Shangqiu University, Shangqiu 476000, China; 6Physical Education College, Shangqiu Normal University, Shangqiu 476000, China

**Keywords:** college student, exercise behavior, negative emotion, mental health, health promotion, COVID-19

## Abstract

Background: In the period of the COVID-19 pandemic, the level of college students’ physical exercise, the detection rate of negative emotions, and their correlation should attract extensive attention. Therefore, this study aimed to explore the correlation between college students’ physical exercise and negative emotions. Methods: Data were collected via a web-based cross-sectional survey. A questionnaire survey was conducted among 3118 college students from five universities in Shanghai in March 2022. In addition to sociodemographic information, measures included Physical Activity Rating Scale (PARS-3) and Depression Anxiety Stress Scale (DASS). The chi-squared test and logistic regression were used to analyze the differences and test the relative risk of negative emotions caused by different amounts of physical exercise. Results: Most students (66.1%) performed a small amount of physical exercise. Male students’ physical-exercise level was higher than female students’, and the detection rate of negative emotions was lower than that of female students. Moderate and low physical-exercise levels were associated with a higher risk of depression (beta of 0.289 and 0.345, respectively) and anxiety (beta of 0.301 and 0.418) symptoms than high physical-exercise level. Conclusions: The anxiety symptoms of college students were significant during the COVID-19 pandemic period. The physical-exercise behavior of college students was closely related to negative emotions, and the weakening of physical-exercise behavior was one of the factors that induced negative emotions in college students.

## 1. Introduction

The COVID-19 pandemic has not only brought about the risk of death from viral infections but also created major public health problems globally, such as plummeting physical activity and outbursts of negative emotions [1,2,3,4,5,6,7]. College students, as a high-risk group for poor mental health during the COVID-19 outbreak, should continue to pay attention to their mental health in order to detect problems as soon as possible and carry out psychological intervention. College students are in the transition stage from adolescence to adulthood. Due to factors such as the control measures taken against COVID-19 and the influence of their own poor cognition, psychological problems have frequently occurred in college students in recent years [8,9,10,11].

College students are prone to emotional and behavioral problems caused by negative emotions [12]. Negative emotions are experiences in which individuals perceive various kinds of pain and unpleasantness, and the production of negative emotions often leads to more intense behaviors (such as suicidal behavior) and physiological reactions (such as crying) [13]. From the classification of negative emotions, depression, anxiety, and stress are the three most common negative emotions among college students [14]. Previous research has shown that these three emotions are interconnected; therefore, they often occur concurrently [15].

To curb the spread of COVID-19, most universities in China have implemented policies to reduce the regional flow of students, requiring college students to reduce unnecessary movement as much as possible, and some universities have implemented the “online classroom” teaching method. Previous studies have shown that the dramatic changes in students’ living and learning styles during major public health emergencies may increase the mental health burden of college students and increase the incidence of negative emotions such as anxiety and depression [1,5,6,16].

Physical activity may be a valuable tool for helping students to maintain their mental health during quarantine. For example, Chen suggested that staying active and performing regular physical activity (primarily recreational physical activity) may help students to recover from mental health issues experienced during isolation during the COVID-19 crisis [17]. The World Health Organization has also published guidance for people in self-isolation, including practical advice on how to stay physically active at home and reduce sedentary behaviors [18]. However, although some studies have explored the association between physical exercise and negative emotions in students from the perspective of public health [7], there are few reports on the association between physical exercise and negative emotions in college students against the background of the COVID-19 pandemic.

In recent years, research related to exercise and health promotion has developed rapidly, especially research on the promotion of physical and mental health from the perspective of behavioral change [19]. However, the topics that scholars pay attention to and the paradigms and methods adopted are not consistent, and the phenomenon of uncertain, inconsistent, and even contradictory research conclusions still exists. This study aims to assess the current status of negative emotions during the COVID-19 pandemic through a large-scale questionnaire survey and to explore the relationship between physical-exercise behavior and negative emotions. This study provides an empirical basis for the mental health management of college students during the normalization period of pandemic prevention and control and the post-pandemic period.

## 2. Materials and Methods

### 2.1. Participants and Procedure

Using the method of stratified random cluster sampling, a questionnaire survey was conducted on freshmen, sophomores, and juniors at five universities in Shanghai in March 2022. Considering that the senior year is the stage of the implementation of educational practice and graduation thesis (design), this study did not sample seniors. According to the administrative class, Questionnaire Star software was used to administer the electronic questionnaires. A total of 3299 questionnaires were distributed, and 3118 valid questionnaires were obtained; the effective rate was 94.5%. The study protocol was approved by the ethics committee at Nantong University (NTU-ISS-2019-001).

The calculation of the minimum sample size was completed using Formula (1) [20], where type I error *α* is set to 0.05, the allowable error *δ* is set to 0.01, and the sample rate *ρ* is set to 0.05. Given a total of 55,018 people (data updated in 2020), the limited overall number *N* sets 75% of the total number of students to be about 41,264. The minimum sample size required for this study was calculated to be 1755.
(1)$$n=\frac{{\left(\frac{Z\alpha }{\delta }\right)}_{}^{2}*p*(1-p)}{1+[{\left(\frac{Z\alpha }{\delta }\right)}_{}^{2}*p*(1-p)]/N}$$

### 2.2. Measurements

The sociological demographic information of college students included sex, grade, age, and school information. Physical-exercise behavior and negative-emotion evaluation were mainly measured using scales as explained below.

#### 2.2.1. Physical Activity Rating Scale (PARS-3)

PARS-3 was compiled by Japanese scholar Hashimoto Kyo and revised by Chinese scholars Liang et al. [21]. PARS-3 calculates the amount of physical exercise from three perspectives—intensity, frequency, and time of physical exercise—and uses this to measure the level of physical exercise in adults. The score is calculated as shown below (2).
***Physical exercise volume score*****=*****intensity*****×** (***time* − 1**) **× *frequency***(2)

Each parameter was evaluated using five score levels. The level standards were as follows: small exercise amount ≤19 points, moderate exercise amount = 20–42 points, and large exercise amount ≥43 points. Follow-up related research showed that the retest reliability of this scale was 0.820, and the internal consistency reliability of PARS-3 was Cronbach’s *α* = 0.85 [22]. The value of the scale was used to reflect the physical-exercise behaviors of college students [22].

#### 2.2.2. Depression Anxiety Stress Scales (DASSs)

The internal structure of DASSs is based on the three-dimensional model proposed by Clark and Watson that argues that depression and anxiety have both unique and common symptom characteristics [23]. In our study, we used *21-Item Depression Anxiety and* Stress (**DASS-C21**), compiled by Lovibond et al., revised by Antony et al. The scale uses the Likert 4-point scoring standard, with scores ranging from 0 to 3, with higher scores indicating higher levels of negative emotions. Wen Yi et al.’s research study found that the reliability and validity of DASS-C21 among Chinese college students were relatively high. The overall Cronbach’s alpha coefficient and test–retest reliability of the scale were 0.912 and 0.751, respectively, and the average correlation coefficient between items was 0.338. The Pearson correlation coefficient of the total score of the scale was 0.895–0.910, and the correlation coefficient between the scores of each subscale was 0.708–0.741 (*p* < 0.01). CFI = 0.914, IFI = 0.912, TLI = 0.893, RMSEA = 0.061 [24].

The classification of DASS-C21 scores in Chinese college students was as follows: A depression score ≤9 was normal; in the range of 10–13, mild; in the range of 14–20, moderate; in the range of 21–27, severe; and ≥28, very severe. An anxiety score ≤7 was considered normal; in the range of 8–9, mild; in the range of 10–14, moderate; in the range of 15–19, severe; and ≥20, very severe. A stress score of ≤14 was considered normal; in the range from 15 to 18, mild; in the range from 19 to 25, moderate; in the range from 26 to 33, severe; and ≥34, very severe [24]. This norm was used to divide the degree of negative emotional symptoms in this study, and the detection rate of negative emotional symptoms was statistically analyzed as the sum of the degrees of mild and above.

### 2.3. Statistical Analysis

All statistical analyses were performed with SPSS 25.0 (IBM Corp., Armonk, NY, USA). The data were first preprocessed, and the missing data were retested (*n* = 34) or eliminated (*n* = 69). The chi-squared test was used to analyze the negative emotional variables in demographic variables and exercise levels. Cramer’s V was used to calculate effect sizes for differences among groups [25]. The ordinal logistic regression analysis model was used to test the relative risk of different physical-exercise levels of negative emotions. The logistic regression analysis model is a computational method for generalized linear models. In the specific calculation, the negative emotion of college students is used as the dependent variable, and the physical-exercise level is used as the factor. Sex and grade were used as covariates. Inspection level *α* = 0.05.

### 2.4. Quality Control

The quality control of the research study included the following aspects: the unification of the research plan and the implementation of the questionnaire survey, special training for the investigators in the early stage of the formal survey, the development of standardized introduction language, the proficiency of the content of the questionnaire, and the correctness of the questionnaire filling. The investigators were college counselors. In the data preprocessing, data such as logical errors and omissions were retested or eliminated to ensure the authenticity and validity of the data. Statistical data-processing requirements were strictly followed; corresponding statistical methods were used for different types of data; and parallelism tests were performed before the orderly logistic regression analysis was performed. A common method bias test was performed before data analysis was applied.

## 3. Results

### 3.1. Status Quo of College Students’ Physical-Exercise Behavior and Detection Rate of Negative Emotions

Overall (see Table 1), the physical-exercise habits of college students were dominated by a small amount of exercise (proportion: 66.1%). In terms of negative emotions, the detection rate of depression was 35.68%, of which severe depression and above accounted for 1.7%; the detection rate of anxiety was 65.5%, of which severe anxiety and above accounted for 7.5%; the detection rate of stress was 11.0%, with severe stress and above accounting for 0.4%. In contrast, the anxiety detection rate was higher. In the difference test, the overall exercise level of men was higher than that of women (*χ*^2^ = 431.912, *p* < 0.001, Cramer’s V = 0.372), and the exercise level of first-grade students was higher than those of second- and third-grade students. For negative emotions, there were differences in depression and anxiety between males and females, with males doing higher than females, but there were no differences in stress between males and females (*χ*^2^ = 16.862, *p* = 0.061). Students in different grades also showed differences in negative emotions (*p* < 0.001).

Table 2 shows an analysis of the differences between the negative emotions of college students performing different amounts of physical exercise. There was a difference in the rate of depression detection among students with different levels of physical exercise (*χ*^2^ = 33.201, *p* < 0.001, Cramer’s V = 0.051), with the normal rate of depression detection being significantly lower for small (62.71%) and medium (63.93%) levels of exercise than for large (70.74%) levels of exercise. There was a difference in the rate of anxiety detection among students with different levels of physical exercise (*χ*^2^ = 16.083, *p* = 0.040, Cramer’s V = 0.073), with the normal rate of anxiety detection being significantly lower for small (32.54%) and medium (32.06%) levels of exercise than for large (43.81%) levels of exercise. No differences in stressful mood were found (*χ*^2^ = 3.889, *p* = 0.69). In terms of sex, it was clear that male students had a greater degree of variation in depression (*χ*^2^ = 18.604, *p* = 0.017, Cramer’s V = 0.084) and anxiety (*χ*^2^ = 16.209, *p* = 0.039, Cramer’s V = 0.079).

### 3.2. The Relationship between College Students’ Physical-Exercise Behavior and Negative Emotions

After controlling for the sex and grade variables, an ordered logistic regression analysis was performed with depression, anxiety, and stress as dependent variables, and the models were tested for parallelism before the analysis, with all of them passing the test (*p* > 0.05). Table 3 showed that the risk of detection of two negative emotions, depression and anxiety, was increased in the little- and moderate-exercise groups compared with the large-amount-of-exercise group (*β* values of 0.348, 0.289, 0.301, and 0.418, respectively; *p* < 0.05); students with a small amount of exercise were 1.16~1.74 times more likely to be depressed and 1.33~1.88 times more likely to be anxious. There were no significant statistical relationships between the occurrence of stress and the amount of exercise (*p* > 0.05).

## 4. Discussion

As the COVID-19 pandemic continues, most college students are required to self-isolate at school or at home, and colleges and universities are implementing online teaching. Therefore, there is strong practical significance in paying attention to college students’ physical-exercise behavior and negative emotions. This study aimed to assess the current status of college students’ physical-exercise behavior and negative emotions during the COVID-19 outbreak and to explore the relationship between the two. The purpose was to provide an empirical basis for the mental health management of college students in the normalization period of pandemic prevention and control and the post-pandemic period. The results of the study showed that during the COVID-19 pandemic, college students’ physical exercise was dominated by a small amount of exercise (66.1%), followed by a large amount of exercise (17.9%); in terms of the detection of negative emotions, anxiety symptoms were significantly high (detection rate 65.5%). The weakening of physical-exercise behavior may be one of the factors that induce negative emotions in college students.

### 4.1. The Physical Exercise of College Students Was Mainly Based on a Small Amount of Exercise during the COVID-19 Outbreak

The results of this study showed that the physical exercise of college students was mainly based on a small amount of exercise (account: 66.1%) and that the exercise level of male students was higher than that of female students. Overall, during the COVID-19 epidemic, college students’ physical exercise was mainly represented by a small amount of exercise, and the level of college students’ participation in physical exercise was worrying, which is similar to the results of previous studies [26,27]. At the same time, this study compared the relevant studies on the physical-exercise behavior of college students using PARS-3 before the outbreak of the COVID-19 and found that the exercise level of college students in this study was weaker than that of previous studies, especially female college students, regardless of the score of PARS-3 or the classification of different exercise levels [28,29,30]. This shows that the impact of the COVID-19 on college students’ physical exercise was significant. At present, the Chinese government adopts the general policy of “dynamic clearing” for the prevention and control of the pandemic. The main pandemic prevention strategy is to reduce the flow of people as much as possible and block the spread of the virus. Therefore, most of the outdoor sports of college students cannot be performed during this time, which is one of the main reasons why the physical-exercise levels in this study were dominated by a small amount of exercise.

From the epidemiological perspective of physical exercise, the physical-exercise level of college students dropped significantly during the COVID-19 pandemic from a global perspective [31]. On the one hand, in recent years, the amount of moderate–high-intensity physical activity in various groups of people around the world declined, which is closely related to changes in nutrition, transportation, and lifestyle [18]. The COVID-19 pandemic exacerbated the decline in physical activity of various groups of people, and the impact of relevant pandemic prevention policies restricted students’ outdoor physical exercise to a certain extent. A study by Alejandro et al. systematically reviewed the impact of the occurrence of COVID-19 on the physical activity of college students and found that during the pandemic, the walking, and moderate, vigorous, and total physical activity levels of college students in different countries significantly decreased [31]. In terms of sex differences, we found that men were more physically active than women, which is consistent with previous studies [32,33]. College students are in a period of transition from adolescence to adulthood. Some studies found that male students and female students showed an upward trend in moderate–high-intensity physical activity before the age of 12 but then show a continuous downward trend, which may be related to the rapid development of human physiology and psychology during adolescence [34]. The results of the eighth national student physique and health survey released by the Chinese Ministry of Education in 2021 showed that the rate of college students’ physical health compliance was still not improved and that the failure rate was as high as 30%. This trend confirms the findings of this study [35].

### 4.2. College Students Had a High Incidence of Anxiety Symptoms during the COVID-19 Outbreak

The results of this study showed that in terms of negative emotions, the detection rate of depressive symptoms was 35.7%; the detection rate of anxiety symptoms was 65.5%; and the detection rate of stress symptoms was 11.0%. Previous studies showed that the detection rate of depressive symptoms among college students in China is between 8% and 74% [36,37,38,39,40]. The detection rate of anxiety symptoms is generally between 5% and 40% [37,38,39]. The detection rate of stress symptoms varies greatly, and there is no fixed detection rate range at present [41]. Xiang’s study used Self-Rating Depression Scale (SDS) and Self-Rating Anxiety Scale (SAS) to measure the negative emotions of 1396 college students in China during the pandemic and found that depression and anxiety symptoms were present in 41.8% and 31.0%, respectively [1]. Ma’s study used Patient Health Questionnaire-9 (PHQ-9) and the Generalized Anxiety Disorder-7 (GAD-7) scale to conduct an online questionnaire survey of 746,217 college students during the pandemic, and the results showed that the detection rates of depression and anxiety symptoms were 21.1% and 11.0%, respectively [6]. Xiao conducted an online survey of 3951 college students using PHQ-9 and GAD-7 9 months after the outbreak in China and found that the prevalence rates of depression and anxiety symptoms among Chinese college students during the pandemic were 59.4% and 54.3% [42].

There are great differences in the detection rates of negative emotions among college students in different studies, which may be related to the cultural backgrounds of the studies, the research paradigms adopted by the researchers, and the differences in measurement tools [37,43]. In addition, the differences in the data may also be related to the pandemic prevention and control policies in the places where the subjects live, the virus transmission intensity, and the media public-opinion environment. Further empirical testing is needed.

There were differences in depression and anxiety among college students of different sexes, with men scoring higher than women, while there were no differences in stress levels between men and women (*χ*^2^ = 16.862, *p* = 0.061). The differences in negative emotions between students of different sexes may be related to factors such as male and female students’ physiology, reaction to negative events, and sensitivity to life events [44]. A study by Vuelvas-Olmos showed that the impact of the COVID-19 pandemic on male and female students was different. Female students reported more psychological distress (such as anxiety and stress), while male students reported higher aggression [44]. There were differences in depression and anxiety among college students of different grades, with the exercise level of first-grade students being higher than those of second- and third-grade students. This may be because senior students face greater academic pressure, resulting in increased psychological pressure.

The detection rate of depressive symptoms in this study was within the range of existing research, but attention should be paid to college students with severe and above detection of depressive symptoms (accounting for 1.7%). This study was implemented during the COVID-19 pandemic, and some studies showed that the incidence of negative emotions such as anxiety and depression in the public increases under major public health emergencies such as COVID-19 [45]. The high incidence of individuals’ negative emotions during major public emergencies is related to a variety of epidemiological and social psychological factors, such as the health of individuals and family members, media exposure, social support, and policy implementation [6]. Lower anxiety levels and higher PE volume levels are more desirable for the university student’s population. Related follow-up studies should increase research of negative emotions such as anxiety and depression, and targeted measures should then be taken to intervene in the mental health management of college students according to the corresponding research results.

### 4.3. The Weakening of Physical-Exercise Behavior Was One of the Factors That Induced the Negative Emotions of College Students during the COVID-19 Outbreak

The results of this study also showed that the detection rate of depression and anxiety among college students with a high level of physical-exercise behavior was lower than that of those with a low level of physical-exercise behavior. At the same time, from the ordinal logistic regression analysis results of negative emotions and physical-exercise behavior, the risk of depression and anxiety detection rates of college students with moderate and low physical-exercise levels were higher than those of students with high physical-exercise levels, which is also consistent with previous research [46,47,48,49].

Previous studies mostly explored the relationship between physical exercise and negative emotions in college students during the COVID-19 pandemic from the perspective of public health. Xiang’s study analyzed the relationship between negative emotions and physical activity among college students during the COVID-19 pandemic and found that high-intensity physical activity (*β* = −0.121, *p* < 0.001) was significantly associated with low anxiety, while moderate- (*β* = −0.095, *p* = 0.001) and high-intensity (*β* = −0.179, *p* < 0.001) physical activity were significantly and strongly associated with a reduction in depression after adjusting for confounding demographic factors [1]. There was a study that showed that the sedentary time (h/day) of college students during the COVID-19 blockade in the United States was significantly correlated with the severity of depression (*β* = 0.29, *p* < 0.05, 95% CI = 0.18, 0.41) and anxiety (*β* = 0.24, *p* < 0.05, 95% CI = 0.13, 0.34) [46]. During the COVID-19 lockdown, leisure physical activity (e.g., stretching, self-resistance training) at home predominated [1]. The results were similar to those of the present study, although the measurement methods were different. The weakening of physical-exercise behavior is one of the factors that induced negative emotions in college students.

The results of this study also provide empirical support for the idea that exercise promotes health. Some studies showed that the reinforcement of physical-exercise behavior may be one of the factors that reduce the occurrence of negative emotions in college students, which is also the theoretical starting point of most current studies on the promotion of mental health via exercise [50,51]. Noora’s research showed that participating in any available exercise or physical activity during the COVID-19 lockdown is important for a person’s health [52]. In general, movement is beneficial, and physical exercise plays an important role in improving the mental health of college students. Follow-up related research such as prospective cohort studies needs to be conducted to further explore the dose–effect relationship of different forms of physical-exercise behaviors in promoting college students’ mental health.

### 4.4. Limitation

The main limitation of this study was that it adopted a cross-sectional study design, so the dose–effect relationship between physical-exercise behavior and negative emotions was not explored. Follow-up prospective cohort studies are needed to further explore the dose–response relationship between the two. In addition, there are many factors that induce negative emotions (such as diet, sleep, home environment, smoking, and drinking) [42,53]. Our study design was a cross-sectional study design, so we only considered physical activity.

## 5. Conclusions

This study found that during the period of COVID-19, college students performed less physical exercise and showed a higher detection rate of negative emotions. The occurrence of negative emotions such as depression and anxiety was closely related to the weakening of physical-exercise behavior, especially the lack of physical exercise and the high incidence of anxiety symptoms among female and senior students. Therefore, during the COVID-19 period, colleges and universities need to pay attention to the physical exercise of college students and the psychological problems caused by negative emotions, formulate relevant strategies and methods to increase the amount of physical exercise among college students, and regularly detect students’ mental health problems with scientific measurement methods.

## Figures and Tables

**Table 1 ijerph-19-10344-t001:** Status sex and grade differences of college students’ negative emotions and physical exercise.

	Overall (*n* = 3118)		Male (*n* = 1305)		Female (*n* = 1813)		First Grade (*n* = 1531)		Second Grade (*n* = 1086)		Third Grade (*n* = 501)	
	*n*	%	*n*	%	*n*	%	*n*	%	*n*	%	*n*	%
Physical-exercise volume score												
Low	2062	66.1	603	46.2	1459	80.5	1124	73.4	626	57.6	312	62.3
Medium	499	16.0	285	21.8	214	11.8	219	14.3	194	17.9	86	17.2
High	557	17.9	417	32.0	140	7.7	188	12.3	266	24.5	103	20.6
*χ* ^2^			431.912				86.260					
** *p* **			<0.001				<0.001					
**Cramer’s V**			0.372				0.118					
Depression												
Normal	2006	64.3	869	66.6	1137	62.7	1041	68.0	644	59.3	321	64.1
Mild	698	22.4	247	18.9	451	24.9	340	22.2	245	22.6	113	22.6
Moderate	360	11.5	157	12.0	203	11.2	139	9.1	161	14.8	60	12.0
Severe	43	1.4	23	1.8	20	1.1	11	0.7	28	2.6	4	0.8
Very Serious	11	0.4	9	0.7	2	0.1			8	0.7	3	0.6
*χ* ^2^			23.841				53.963					
** *p* **			<0.001				<0.001					
**Cramer’s V**			0.087				0.093					
Anxiety												
Normal	1075	34.5	521	39.9	554	30.6	526	34.4	369	34.0	180	35.9
Mild	855	27.4	328	25.1	527	29.1	450	29.4	267	24.6	138	27.5
Moderate	954	30.6	351	26.9	603	33.3	456	29.8	350	32.2	148	29.5
Severe	178	5.7	73	5.6	105	5.8	87	5.7	62	5.7	29	5.8
Very Serious	56	1.8	32	2.5	24	1.3	12	0.8	38	3.5	6	1.2
*χ* ^2^			39.064				34.482					
** *p* **			<0.001				<0.001					
**Cramer’s V**			0.112				0.074					
Stress												
Normal	2775	89.0	1157	88.7	1618	89.2	1399	91.4	931	85.7	445	88.8
Mild	249	8.0	94	7.2	155	8.6	107	7.0	105	9.7	37	7.4
Moderate	83	2.7	44	3.4	39	2.2	25	1.6	42	3.9	16	3.2
Severe	11	0.4	10	0.8	1	0.1	0.00	0.0	8	0.7	3	0.6
Very Serious	0	0.0	0	0.0	0	0.0	0	0.0	0	0.0	0	0.0
*χ* ^2^			16.862				31.613					
** *p* **			0.061				<0.001					
**Cramer’s V**			0.074				0.071					

**Table 2 ijerph-19-10344-t002:** Differences in negative emotions of college students for different physical-exercise quantities.

		Overall (*n* = 3118)						Male (*n* = 1305)						Female (*n* = 1813)					
		Low		Medium		High		Low		Medium		High		Low		Medium		High	
		(*n* = 2062)		(*n* = 499)		(*n* = 557)		(*n* = 603)		(*n* = 285)		(*n* = 417)		(*n* = 1459)		(*n* = 214)		(*n* = 140)	
		*n*	%	*n*	%	*n*	%	*n*	%	*n*	%	*n*	%	*n*	%	*n*	%	*n*	%
**Depression**																			
	Normal	1293	62.7	319	63.9	394	70.7	383	63.5	185	64.9	301	72.2	910	62.4	134	62.6	93	66.4
	Mild	480	23.3	115	23.1	103	18.5	111	18.4	61	21.4	75	18.0	369	25.3	54	25.2	28	20.0
	Moderate	253	12.3	57	11.4	50	9.0	91	15.1	32	11.2	34	8.2	162	11.1	25	11.7	16	11.4
	Severe	31	1.5	5	1	7	1.3	15	2.5	4	1.4	4	1.0	16	1.1	1	0.5	3	2.1
	Very Serious	5	0.2	3	0.6	3	0.5	3	0.5	3	1.1	3	0.7	2	0.1	0	0	0	0
	*χ* ^2^	33.201						18.604						4.491					
	** *p* **	<0.001						0.017						0.81					
	**Cramer’s V**	0.051						0.084						0.035					
Anxiety																			
	Normal	671	32.5	160	32.1	244	43.8	230	38.1	101	35.4	190	45.6	441	30.2	59	27.6	54	38.6
	Mild	565	27.4	141	28.3	149	26.8	141	23.4	75	26.3	112	26.9	424	29.1	66	30.8	37	26.4
	Moderate	672	32.6	153	30.7	129	23.2	178	29.5	84	29.5	89	21.3	494	33.9	69	32.2	40	28.6
	Severe	118	5.7	35	7.0	25	4.5	38	6.3	17	6.0	18	4.3	80	5.5	18	8.4	7	5.0
	Very Serious	36	1.8	10	2	10	1.8	16	2.7	8	2.8	8	1.9	20	1.4	2	0.9	2	1.4
	*χ* ^2^	16.083						16.209						8.544					
	** *p* **	0.04						0.039						0.382					
	**Cramer’s V**	0.073						0.079						0.049					
Stress																			
	Normal	1831	88.8	441	88.4	503	90.3	527	87.4	250	87.7	380	91.1	1304	89.4	191	89.3	123	87.9
	Mild	170	8.2	42	8.4	37	6.6	47	7.8	23	8.1	24	5.8	123	8.4	19	8.9	13	9.3
	Moderate	56	2.7	13	2.6	14	2.5	25	4.2	9	3.2	10	2.4	31	2.1	4	1.9	4	2.9
	Severe	5	0.2	3	0.6	3	0.5	4	0.7	3	1.1	3	0.7	1	0.1	0	0	0	0
	Very Serious	0	0	0	0	0	0	0	0	0	0	0	0	0	0	0	0	0	0
	*χ* ^2^	3.889						4.91						0.824					
	** *p* **	0.69						0.555						0.991					
	**Cramer’s V**	0.025						0.043						0.015					

**Table 3 ijerph-19-10344-t003:** Ordinal logistic regression analysis of negative emotions and physical-exercise volume score (*n* = 3118).

		Low	Medium
**Depression**			
	** *β* **	0.348	0.289
	Standard error	0.109	0.130
	Wald *χ*^2^	11.934	5.179
	OR	1.421	1.338
	95% CI	1.163~1.742	1.038~1.728
	** *p* **	0.001	0.023
Anxiety			
	** *β* **	0.301	0.418
	Standard error	0.884	0.114
	Wald *χ*^2^	27.121	18.090
	OR	1.581	1.159
	95% CI	1.325~1.884	1.301~2.020
	** *p* **	<0.001	<0.001
Stress			
	** *β* **	0.155	0.204
	Standard error	0.061	0.119
	Wald *χ*^2^	0.951	0.990
	OR	1.166	1.219
	95% CI	−0.860~1.591	−0.831~1.800
	** *p* **	0.331	0.323

## Data Availability

The raw data supporting the conclusions of this article can be made available by the authors, without undue reservation.

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
