# Peer review of "Chinese College Students’ Physical-Exercise Behavior, Negative Emotions, and Their Correlation during the COVID-19 Outbreak"

_ijerph, 2022, doi:10.3390/ijerph191610344_

Round 1

Reviewer 1 Report

I want to thank the authors and the Editorial Board for the opportunity to review the article submitted to the International Journal of Environmental Research and Public Health. The authors’ manuscript refers to a very important topic: students’ functioning during the COVID-19 outbreak. All statistical methods were applied correctly and the study has sufficient theoretical background. I believe that the submitted article is of high quality. I recommend the publication of the authors’ manuscript after one major change.

I highly recommend that authors supplement all Tables (1, 2 and 3) with proper effect size measures. P-values are related to the sample size (see Lakens, 2022). In big samples, such as this presented in the authors’ manuscript, even small (marginal) effects can come out as significant. Therefore it would be better to discuss all results in relation to the effect size rather than statistical significance. It is extremely important. From the practical point of view, should we be interested in very small, almost non-existent effects, only because they came out as significant?

Additionally, I recommend that authors report all p-values with 3 decimal places. I also recommend that the authors report model fit measures such as CFI and RMSEA also with 3 decimal places, especially for DASS questionnaire.

Author Response

Point-by-point Responses to Reviewer 1

Dear reviewer,

Thank you for the time and effort that you have dedicated to providing your insightful and valuable comments on our manuscript. Although I do not know the situation around you, please stay healthy and keep safe. Here are point-by-point responses to your comments, I hope the responses address your concerns effectively.

Sincerely,

Comment 1:

I highly recommend that authors supplement all Tables (1, 2 and 3) with proper effect size measures. P-values are related to the sample size (see Laken’s, 2022). In big samples, such as this presented in the authors’ manuscript, even small (marginal) effects can come out as significant. Therefore, it would be better to discuss all results in relation to the effect size rather than statistical significance. It is extremely important. From the practical point of view, should we be interested in very small, almost non-existent effects, only because they came out as significant?

Response 1:

Thank you for your comment. This is a very good suggestion. We all accept your suggestions and described in the Methods and Results sections of the text. The Cramer's V coefficient was used to calculate the effect size.

[1] Cohen, J. (1988). Statistical Power Analysis for the Behavioral Sciences (2nd ed.). Hillsdale, NJ: Lawrence Erlbaum Associates, Publishers.

Comment 2:

I recommend that authors report all p-values with 3 decimal places. I also recommend that the authors report model fit measures such as CFI and RMSEA also with 3 decimal places, especially for DASS questionnaire.

Response 2:

Thank you for your comment. This is also a very good suggestion. We all accept your suggestions.

Reviewer 2 Report

General comments:

Thank you for submitting this interesting article. Let me positively highlight the idea to study the Chinese college students' physical exercise behavior, and negative emotions.

In general, my opinion is that before the publication of this interesting study, some important questions should be (re)considered. I will explain it in the following text.

Introduction

Introduction it was well written and the background of this study was quite well described and explained.   

Methods

In general, methods were very well chosen and described, and in detailed manner.

In the methods, it was not explained how test-retest reliability was determined, i.e. in a follow-up measurement after one month with how many students involved. That also means that you somehow did match-up data for every student involved in these two measurements with your Questionnaire Star software (please, name the author/-s of that software into the article).

Additionally, in the methods (and in the article), it was written three different numbers of involved subjects (3299; 3187; 3118), and it was calculated that effective rate is 97.8%. Shouldn’t the effective rate be calculated accordingly to the overall number of involved students (3299), and then the effective rate would be 94.5%. The number 3299 was the ‘starting’ point, but only 3118 students were analyzed, and in abstract could be mentioned only 3118 subjects (because the study obtained results relied only and particularly on them).

Results

Into the results there are some issues that are needed to be clarified and (re)considered additionally.

Table 1. - Differences by gender and by the degree of study were qualitatively determined. My recommendation is to use Cramer's V coefficient to determine precisely effect size of the previously determined positive chi-square test. There are some different explanations and different criteria of interpretation of the obtained Cramer's V coefficient (e.g., by Cohen, 1988, …).

Table 2. – It is solidly made, but before reaching any conclusion regarding these results and this table, the following question should be considered in detail.

If differences were determined previously by the gender of the students [especially for the physical exercise (PE) level, but also for the depression (D) and anxiety (A) levels, and for stress (SS) it was very close to significance], was it necessary to analyze and to represent the results from Table 2. separately for the female and male subsamples of students? There were quite enough of them for such an analysis (Males n - 1305; Females n - 1813). Furthermore, according to the measured characteristics (PE, D and A), these samples of subjects do not belong to the same population (because significant differences were found) and it was necessary to analyze them separately! If we analyze them together and overall, then the results of analyzes from Table 2 can be "masked" by previously established gender differences – as well, then the conclusions of this study could lose their overall value! Example: for the low level of PE in the overall sample the percentage was 66.1%, and it included the calculated mean of two measures, of 46.2% for the Males subsample and of 80.5% for the Females subsample. These are very large and significant differences, which should be considered in detail. Thus, for the two remaining PE levels (medium PE level: M - 21.8%, F - 11.8%; high PE level, M - 32.0%, F - 7.7%, even 4 times more males students were found in the high PE level than females students). If you accept my suggestion, then Table 2 should have two parts, especially for male and for female respondents.

As for the also established differences by the degree of study, although the differences between the three degrees of study were determined, that three subsamples should not be analyzed separately later. Rather, my opinion is that you could only try to establish a clear 'temporal' pattern or some form of temporal connection of the degree of the study with the measured characteristics (PE, D, A and SS). I have seen ‘increasing’ of the PE level at the 2nd and 3rd year of study compared to the 1st year students in the overall sample (from the Table 1). Additionally, I didn’t find the interpretation of that findings into the discussion section except that sentence into the lines 171 - 172.

Table 3. – Why wasn't multiple regression done on high PE level volume respondents (n=557) to determine their structure of negative emotions, as well! And that could be a possible significant finding for the future! That differences between three PE volume level groups could show interesting empirical findings for future and for possible PE policy recommendations (for actions or decisions), and for promoting health on the universities.

Discussion and Conclusions

The discussion and conclusions sections were made good and in a meaningful manner, accordingly to the previously presented results.

Specific comments:

- Abstract - My opinion is that the abstract is needed to be changed and to be shortened. Your abstract is 322 words long (it could be up to 200 words). Example: whole sentence ‘Chi square test was used to …' could be ejected from the abstract. All you want to present in this abstract should be explained within only 200 words (± 10%). As well, it should be divided to four sections named: 1) Background (not as Objective); 2) Methods; 3) Results; and 4) Conclusion. Further, how many subject you had analyzed (3118 or 3299) at all?

- Section 2.2.2.  – Please, try to present the names of the DASS-C21 questionnaire measures in the ITALIC form, as depression, anxiety, or stress. Those are the names of the DASS-C21 measures, and they are very important.

- Line 131 – ‘CFI = 0.91, IFI = 0.91, TLI = 0.89, RMSEA = 0.06 [21].' Is it sentence or quote??? Please, mention that it was found and determined during the validation of the Chinese DAS-21 version by the Wen Yi et al. (2012).

- Lines 218-220, 281-282, and in Abstract –

'… the exercise level of male students is better than that of female students.'   The using of the words BETTER or WORSE in scientific papers is not common, and the expressions HIGHER or LOWER should be used to express the relations between two samples or subsamples. Later in the interpretation (in the discussion section) it could be additionally clarified which characteristic (higher or lower) is more desirable or more suitable (and thus ‘better’) for a subjects population. Example: Lower anxiety level and higher PE volume level are more desirable for the university students population.

- Results - In all tables, consider to use just decimal numbers with just two digits after the decimal point (e.g., χ2 = 3.89, P = 0.69) because it would be easily to read and to compare different results. By my opinion, using of three digits results is appropriate just when you are using them to confirm coefficient effect or validation ‘strength’ (effect size) compared to some already determined criterion or limit (e.g., GFI = 0.983, TLI = 0.982, CFI = 0.984, RMSEA = 0.055; or Cramer’s V = 0.142). As well, all proportions should be written with just one digit behind decimal point (e.g., 66.1%) because it is the widely accepted ‘standard’ in the publishing of scientific studies

Author Response

Point-by-point Responses to Reviewer 2

Dear reviewer,

Thank you for the time and effort that you have dedicated to providing your insightful and valuable comments on our manuscript. Although I do not know the situation around you, please stay healthy and keep safe. Here are point-by-point responses to your comments, I hope the responses address your concerns effectively.

Sincerely,

Comment 1:

In the methods, it was not explained how test-retest reliability was determined, i.e. in a follow-up measurement after one month with how many students involved. That also means that you somehow did match-up data for every student involved in these two measurements with your Questionnaire Star software (please, name the author/-s of that software into the article).

Response 1:

Thank you for your comment. It should be noted that PARS-3 and DASS is a commonly used behavior scale. For this reason, our team tested the applicability of PARS-3 and DASS in Chinese college students in an earlier study, and the results also showed that PARS -3 and DASS is suitable for application among Chinese college students [1,2]. Since the test subjects in this study were also college students, in order to improve the timeliness of the study, the reliability and validity of PARS-3 and DASS were not tested in this study.

[1]. Li, B., et al., Promoting exercise behavior and cardiorespiratory fitness among college students based on the motivation theory. BMC Public Health, 2022. 22(1).

[2]. Wen Yi, et al., Evaluation of the reliability and validity of the Chinese version of the Depression-Anxiety-Stress Scale. Chinese Public Health, 2012. 28(11):1436-1438.

Comment 2:

Additionally, in the methods (and in the article), it was written three different numbers of involved subjects (3299; 3187; 3118), and it was calculated that effective rate is 97.8%. Shouldn’t the effective rate be calculated accordingly to the overall number of involved students (3299), and then the effective rate would be 94.5%. The number 3299 was the ‘starting’ point, but only 3118 students were analyzed, and in abstract could be mentioned only 3118 subjects (because the study obtained results relied only and particularly on them).

Response 2:

Thank you for your comment. This is a very good suggestion. We all accept your suggestions.

Comment 3:

Table 1. - Differences by gender and by the degree of study were qualitatively determined. My recommendation is to use Cramer's V coefficient to determine precisely effect size of the previously determined positive chi-square test. There are some different explanations and different criteria of interpretation of the obtained Cramer's V coefficient (e.g., by Cohen, 1988, …).

Response 3:

Thank you for your comment. This is a very good suggestion. We all accept your suggestions and described in the Methods and Results sections of the text. The Cramer's V coefficient was used to calculate the effect size [1,2].

[1] Cohen, J. (1988). Statistical Power Analysis for the Behavioral Sciences (2nd ed.). Hillsdale, NJ: Lawrence Erlbaum Associates, Publishers.

[2] Michael W. Kearney, Cramér's V, In book: Sage Encyclopedia of Communication Research Methods, Publisher: Sage (https://www.researchgate.net/publication/307963787_Cramer's_V)

Comment 4:

Table 2. – It is solidly made, but before reaching any conclusion regarding these results and this table, the following question should be considered in detail.

If differences were determined previously by the gender of the students [especially for the physical exercise (PE) level, but also for the depression (D) and anxiety (A) levels, and for stress (SS) it was very close to significance], was it necessary to analyze and to represent the results from Table 2. separately for the female and male subsamples of students? There were quite enough of them for such an analysis (Males n - 1305; Females n - 1813). Furthermore, according to the measured characteristics (PE, D and A), these samples of subjects do not belong to the same population (because significant differences were found) and it was necessary to analyze them separately! If we analyze them together and overall, then the results of analyzes from Table 2 can be "masked" by previously established gender differences – as well, then the conclusions of this study could lose their overall value! Example: for the low level of PE in the overall sample the percentage was 66.1%, and it included the calculated mean of two measures, of 46.2% for the Males subsample and of 80.5% for the Females subsample. These are very large and significant differences, which should be considered in detail. Thus, for the two remaining PE levels (medium PE level: M - 21.8%, F - 11.8%; high PE level, M - 32.0%, F - 7.7%, even 4 times more males students were found in the high PE level than females students). If you accept my suggestion, then Table 2 should have two parts, especially for male and for female respondents.

As for the also established differences by the degree of study, although the differences between the three degrees of study were determined, that three subsamples should not be analyzed separately later. Rather, my opinion is that you could only try to establish a clear 'temporal' pattern or some form of temporal connection of the degree of the study with the measured characteristics (PE, D, A and SS). I have seen ‘increasing’ of the PE level at the 2nd and 3rd year of study compared to the 1st year students in the overall sample (from the Table 1). Additionally, I didn’t find the interpretation of that findings into the discussion section except that sentence into the lines 171 - 172.

Response 4:

Thank you for your comment. As you say, not doing an analysis of gender differences does overwhelm some "significant" findings. The result is as you expected. We fully accept your suggestion and have made a comprehensive revision to Table 2.

Comment 5:

Table 3. – Why wasn't multiple regression done on high PE level volume respondents (n=557) to determine their structure of negative emotions, as well! And that could be a possible significant finding for the future! That differences between three PE volume level groups could show interesting empirical findings for future and for possible PE policy recommendations (for actions or decisions), and for promoting health on the universities.

Response 5:

Thank you for your comment. The ordinal logistic regression analysis is a standardized statistical analysis method. Logistic regression analysis model is a computational method for generalized linear models. In the specific calculation, the negative emotion of college students is used as the dependent variable, and the physical activity level is used as the factor. Gender and grade were used as covariates. In the interpretation of the results, high PE level is used as the baseline for comparison.

Comment 6:

The discussion and conclusions sections were made good and in a meaningful manner, accordingly to the previously presented results.

Response 6:

Thank you for your comment. We followed your suggestion and deepened the Discussion section based on the revised results.

Comment 7:

- Abstract - My opinion is that the abstract is needed to be changed and to be shortened. Your abstract is 322 words long (it could be up to 200 words). Example: whole sentence ‘Chi square test was used to …' could be ejected from the abstract. All you want to present in this abstract should be explained within only 200 words (± 10%). As well, it should be divided to four sections named: 1) Background (not as Objective); 2) Methods; 3) Results; and 4) Conclusion. Further, how many subject you had analyzed (3118 or 3299) at all?

Response 7:

Thank you for your comment. We have rewritten the abstract with your suggestion.

Comment 8:

- Section 2.2.2.  – Please, try to present the names of the DASS-C21 questionnaire measures in the ITALIC form, as depression, anxiety, or stress. Those are the names of the DASS-C21 measures, and they are very important.

Response 8:

Thanks for your suggestion, we have made changes in the original, please see 2.2.2.

Comment 9:

- Line 131 – ‘CFI = 0.91, IFI = 0.91, TLI = 0.89, RMSEA = 0.06 [21].' Is it sentence or quote??? Please, mention that it was found and determined during the validation of the Chinese DAS-21 version by the Wen Yi et al. (2012).

Response 9:

Thanks for your suggestion, we have made changes in the original, please see 2.2.2.

Comment 10:

- Lines 218-220, 281-282, and in Abstract –

'… the exercise level of male students is better than that of female students.'  –  The using of the words BETTER or WORSE in scientific papers is not common, and the expressions HIGHER or LOWER should be used to express the relations between two samples or subsamples. Later in the interpretation (in the discussion section) it could be additionally clarified which characteristic (higher or lower) is more desirable or more suitable (and thus ‘better’) for a subjects population. Example: Lower anxiety level and higher PE volume level are more desirable for the university students population.

Response 10:

Thank you very much for your suggestion. We strongly agree with your suggestion and have made changes in the article.

Comment 11:

- Results - In all tables, consider to use just decimal numbers with just two digits after the decimal point (e.g., χ2 = 3.89, P = 0.69) because it would be easily to read and to compare different results. By my opinion, using of three digits results is appropriate just when you are using them to confirm coefficient effect or validation ‘strength’ (effect size) compared to some already determined criterion or limit (e.g., GFI = 0.983, TLI = 0.982, CFI = 0.984, RMSEA = 0.055; or Cramer’s V = 0.142). As well, all proportions should be written with just one digit behind decimal point (e.g., 66.1%) because it is the widely accepted ‘standard’ in the publishing of scientific studies.

Response 11:

Thank you very much for your suggestion. We strongly agree with your suggestion and have made changes in the article.

Reviewer 3 Report

Overall

By definition, the Covid-19 outbreak is classified as a pandemic rather than an epidemic. Perhaps the authors can reconsider their framing, and be consistent throughout   Abstract - Line 19 - Consider ‘were’ after ‘Data’   Introduction - There is considerable room for impairment in the introduction, mention of physical activity among college students is notably absent. Moreover, the way information is presented does not leave me sold that the authors are filling a notable gap in the literature. I’m left wondering a lot when it comes to the what we know and what we don’t know of the study to support the author’s rationale for the what they aim to find our - "Due to factors such as the control measures taken against COVID-19 47 and the influence of their own poor cognition, psychological problems have frequently occurred in college students in recent years [8].” This is a big statement to make that is only supported by a single study conducted in France. Not saying I disagree entirely, but to make such a statement it’d be helpful to better support it   Methods - A response rate >90% is incredibly high, especially in this population. Can the authors offer insight into recruitment methods that help to explain this - What was the rationale for using the PARS-3 over other more widely used  physical activity measure such as the IPAQ and GPAQ that would have allowed for comparisons with previous research. Low, medium and high hold far less utility than simply being able to say, X% of students met aerobic physical activity recommendations - What do the authors mean when referring to ‘retesting’ missing data - Perhaps consider providing some context regarding the circumstances under which data were collected - You say you assessed gender, but what happened to students who were not male or female? - Please offer more details on exactly how regression analyses were conducted   Results - Further context is required regarding first, second, and third grade as such terminology is not common in this setting in many other countries - Table 2 - Is it justifiable to compare differences in mental health based on exercise after having established that there are gender/grade differences without controlling for gender/grade in the analyses? This analyses is seemingly made redundant by the subsequent regression - Table 3 - Please be clear about what the referent group was, and how analyses were conducted, at the moment detail is lacking. Regression analyses do not show ‘increases’, just associations which may require some slight rewording   Discussion - Be specific about the time in the pandemic that this was conducted - It is worth considering not using physical activity and exercise interchangeably, they are different concepts and thus the message varies based on what is used   Conclusion - As you stated, your study is cross-sectional so you cannot conclude that anything changed or decreased. You go on to imply temporal associations between physical activity and mental health, which again you cannot do based on your data. I recommend you reconsider messaging and wording. 

Author Response

Point-by-point Responses to Reviewer 3

Dear reviewer,

Thank you for the time and effort that you have dedicated to providing your insightful and valuable comments on our manuscript. Although I do not know the situation around you, please stay healthy and keep safe. Here are point-by-point responses to your comments, I hope the responses address your concerns effectively.

Sincerely,

Comment 1:

By definition, the Covid-19 outbreak is classified as a pandemic rather than an epidemic. Perhaps the authors can reconsider their framing, and be consistent throughout  

Response 1:

Thank you very much for your comment. We have re-examined and made corrections.

Comment 2:

Abstract - Line 19 - Consider ‘were’ after ‘Data’

Response 2:

Thank you for very much your comment. We have made changes.

Comment 3:

Introduction - There is considerable room for impairment in the introduction, mention of physical activity among college students is notably absent. Moreover, the way information is presented does not leave me sold that the authors are filling a notable gap in the literature. I’m left wondering a lot when it comes to the what we know and what we don’t know of the study to support the author’s rationale for the what they aim to find us.

Response 3:

Thank you for your comment. In order to more clearly show the purpose, significance and research progress of this study at home and abroad. We have rearranged the logical order of A and added some up-to-date literature information.

Comment 4:

Due to factors such as the control measures taken against COVID-19 47 and the influence of their own poor cognition, psychological problems have frequently occurred in college students in recent years [8].” This is a big statement to make that is only supported by a single study conducted in France. Not saying I disagree entirely, but to make such a statement it’d be helpful to better support it  

Response 4:

Thank you for your comment. We very much agree with your suggestion. To enhance the level of evidence for our conclusions, we have added recent relevant studies to validate this claim[1-3]. It can be seen that there are many reports on the incidence of negative emotions among Chinese college students.

  1. Wang C, Tang N, et al. Need for cognitive closure and trust towards government predicting pandemic behavior and mental health: comparing United States and China. Curr Psychol. 2022 Jul 5:1-14.
  2. Shek DTL, Dou D, et al. Need Satisfaction and Depressive Symptoms Among University Students in Hong Kong During the COVID-19 Pandemic: Moderating Effects of Positive Youth Development Attributes. Front Psychiatry. 2022 Jul 7;13:931404.
  3. Peng X, Liu L,et al. Longitudinal changes in fear and anxiety among Chinese college students during the COVID-19 pandemic: a one-year follow-up study. Curr Psychol. 2022 Jul 26:1-10.

Comment 5:

Methods - A response rate >90% is incredibly high, especially in this population. Can the authors offer insight into recruitment methods that help to explain this?

Response 5:

Thank you for your comment. Maintaining a high questionnaire recovery rate (effective questionnaire rate) is a guarantee for us to obtain real and reliable data. Generally speaking, if the recovery rate is only about 30%, the data can only be used as a reference; if the recovery rate is more than 50%, suggestions can be adopted; when the recovery rate reaches more than 70-75%, it can be used as the basis for research conclusions. Therefore, the recovery rate of the questionnaire should generally not be less than 70%[1,2].

  • Willer D , Walker H . Building Experiments: Testing Social Theory. 2007.
  • Federica Russo.Causality and causal modelling in the social sciences New York : Springer, 2008.

Comment 6:

What was the rationale for using the PARS-3 over other more widely used physical activity measure such as the IPAQ and GPAQ that would have allowed for comparisons with previous research? Low, medium and high hold far less utility than simply being able to say, X% of students met aerobic physical activity recommendations.

Response 6:

Thank you for your comment. We apply PARS-3 for two reasons. First, our research tends to focus on physical exercise behavior, which is the so-called recreational physical activity. Second, PARS-3 is a relatively mature and simple physical exercise behavior measurement tool, which is currently used in a wide range of applications.

Comment 7:

What do the authors mean when referring to ‘retesting’ missing data.

Response 7:

Thank you for your comment. What this means here is that data that is missing key information (such as gender, grade, etc.) need to be eliminated when performing specific calculations.

Comment 8:

You say you assessed gender, but what happened to students who were not male or female?

Response 8:

Thank you very much for your comment. This is a very good question. We are impressed by your expertise. In the specific modification, we replaced all gender with sex. Sex puts more emphasis on the biological dichotomy of humans, which corresponds to males and females in this study.

Comment 9:

Please offer more details on exactly how regression analyses were conducted.

Response 9:

Thank you for your comment. This is a very good suggestion. We all accept your suggestions and described in the Methods and Results sections of the text. The Cramer's V coefficient was used to calculate the effect size [1,2].

[1] Cohen, J. (1988). Statistical Power Analysis for the Behavioral Sciences (2nd ed.). Hillsdale, NJ: Lawrence Erlbaum Associates, Publishers.

[2] Michael W. Kearney, Cramér's V, In book: Sage Encyclopedia of Communication Research Methods, Publisher: Sage (https://www.researchgate.net/publication/307963787_Cramer's_V)

Comment 10:

Results - Further context is required regarding first, second, and third grade as such terminology is not common in this setting in many other countries

Response 10:

Thank you for your comment. The academic system of Chinese universities is generally 4 to 5 years to complete university studies. First, because the general professional knowledge is divided into three levels: basic courses, professional basic courses, professional courses, one level per year, plus one year of internship, it is exactly four years. The reason for the five-year undergraduate medical degree is that the "internship" link was added in the fourth year to show that "human life is at stake", and the fifth year is still an internship. Another reason is that the average person does not become fully biologically mature until the age of twenty-two. Some students may feel that their stature is a little lower. It doesn't matter. In the four years of university, not only knowledge but also height is required. Everyone still has four or five years to be long enough. In this way, college students enter college at the age of 18 or 9, and their bodies are just mature after graduating at the age of 22 or 3. It is more appropriate to go to work at this time. In this study, only the first grade, the second grade and the third grade were selected, and the fourth grade was not selected. Mainly because fourth grade students are in the stage of graduation thesis, graduation project or internship.

Comment 11:

Table 2 - Is it justifiable to compare differences in mental health based on exercise after having established that there are gender/grade differences without controlling for gender/grade in the analyses? This analysis is seemingly made redundant by the subsequent regression.

Response 11:

Thank you for your comment. You have raised a very good question. Among the specific revisions, we conducted independent analyses of students of different genders and found some "significant" findings(Please refer to Table 2). These important conclusions are drawn from your pragmatism and professionalism. Thanks again for your comment.

Comment 12:

Table 3 - Please be clear about what the referent group was, and how analyses were conducted, at the moment detail is lacking. Regression analyses do not show ‘increases’, just associations which may require some slight rewording.

Response 12:

Thank you for your comment. We refined it in the Methods section at your suggestion. The ordinal logistic regression analysis model was used to test the relative risk of different physical exercise levels on negative emotions. Logistic regression analysis model is a computational method for generalized linear models. In the specific calculation, the negative emotion of college students is used as the dependent variable, and the physical exercise level is used as the factor. Sex and grade were used as covariates.

Comment 13:

Be specific about the time in the pandemic that this was conducted.

Response 13:

Thank you for your comment. This research using the method of stratified random cluster sampling, a questionnaire survey was conducted on freshmen, sophomores, and juniors at five universities in Shanghai in March 2022. This study used a cross-sectional study design. And a "contactless" online questionnaire was used, so our study was conducted during the COVID-19 pandemic in Shanghai, China.

Comment 14:

It is worth considering not using physical activity and exercise interchangeably, they are different concepts and thus the message varies based on what is used.

Response 14:

Thank you very much for reminding. We re-examined the text and revised where the two terms were confused.

Comment 15:

Conclusion - As you stated, your study is cross-sectional so you cannot conclude that anything changed or decreased. You go on to imply temporal associations between physical activity and mental health, which again you cannot do based on your data. I recommend you reconsider messaging and wording.

Response 15:

Thank you very much for your suggestion. We have rethought the delivery and wording of the message as you suggested.

Round 2

Reviewer 2 Report

Dear manuscript authors,

thank you for provided and detailed resposes on my comments. I'm glad if I was able to help you make your manuscript better.

Additional comment – By my opinion, there is no need to present OVERALL sample results in Table 2 because they „mask“ statistically totally different results obtained in the two different subsamples by gender (by sex), in the measure of depression (males, p=0.02, females p=0.81), and in the measure of anxiety (males, p=0.04, females p=0.38). By my opinion, that overall results do not represent nothing scientifically important, and that is additionally 'confirmed' by the two different and important results presented into the same table. As well, into the IJERPH's guidelines for preparing manuscript is written „To facilitate the copy-editing of larger tables, smaller fonts may be used, but no less than 8 pt. in size.” For me, it looks like that fonts of the presented results in Table 2 are smaller than proposed 8 pt. size.

Best regards for your future personal and professional work!